# The Meyers Estimates for Domains Perforated along the Boundary

**Gregory A. Chechkin** [1,2,3]

---

[1] Department of Differential Equations, Faculty of Mechanics and Mathematics, M.V. Lomonosov Moscow State University, Leninskie Gory, 1, 119991 Moscow, Russia; chechkin@mech.math.msu.su

[2] Institute of Mathematics with Computing Center, Subdivision of the Ufa Federal Research Center of Russian Academy of Science, Chernyshevskogo st., 112, 450008 Ufa, Russia

[3] Institute of Mathematics and Mathematical Modeling, Pushkin st. 125, Almaty 050010, Kazakhstan

**Abstract:** In this paper, we consider an elliptic problem in a domain perforated along the boundary. By setting a homogeneous Dirichlet condition on the boundary of the cavities and a homogeneous Neumann condition on the outer boundary of the domain, we prove higher integrability of the gradient of the solution to the problem.

**Keywords:** higher integrability; Meyers estimates; perforated domain; small parameter

## 1. Introduction

This paper confronts the estimates of solutions to an elliptic problem in domains perforated along the boundary (see Figure 1).

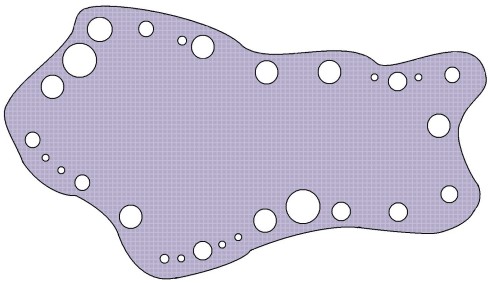

**Figure 1.** Domain perforated along the boundary.

For the following homogeneous Dirichlet problem in a bounded domain:

$$
\begin{cases}
\mathcal{L}u := \operatorname{div}(a(x)\nabla u) = \operatorname{div} f, & x \in \Omega, \\
u = 0, \ x \in \partial\Omega
\end{cases}
\tag{1}
$$

with uniformly elliptic measurable and symmetric matrix $a(x)$, that is, $a_{ij} = a_{ji}$, and the following:

$$
\lambda^{-1}|\xi|^2 \le \sum_{i,j=1}^{d} a_{ij}(x)\xi_i\xi_j \le \lambda|\xi|^2, \text{ for almost all } x \in \Omega, \text{ and for all } \xi \in \mathbb{R}^d,
\tag{2}
$$

and with the right hand side as $f \in L_p(\Omega)$, where $p > 2$, a higher integrability of the gradient of solutions (Meyers estimates) in a plane domain was proved in [1]. In other words, it was proved that the gradient of the solution is integrable at the power greater than two:

$$\int_\Omega |\nabla u_\varepsilon|^{2+\delta} dx \le C \int_\Omega |f|^{2+\delta} \, dx. \tag{3}$$

In a multidimensional case, the same result for domains with sufficiently smooth boundary was proved in [2]. It should be noted that higher integrability of the gradient of solutions to the Dirichlet problem in a bounded domain with Lipschitz boundary for $p$-Laplacian with variable $p$ was obtained in [3].

The Meyers estimate (higher integrability) of solutions to a Zaremba problem with rapidly changing type of boundary conditions in a plane domain for the Laplacian can be observed in [4]. The uniformly elliptic operators in the multidimensional case can be observed in [5].

Some other integral estimates of solutions can be found in [6–9]. In paper [10], one can find the integral estimates in domains perforated along the boundary.

It should be noted that similar mathematical models and problems appear in many applications, for instance, in mechanics of aircraft and space structures, theory of bridge constructions, hudrodynamics of bodies with complicated microstructure, etc. For more details, refer to [11].

This paper is devoted to obtaining the Meyers estimates for the gradient of the solution to an elliptic problem on a perforated slope along the boundary. Thus, by assuming a homogeneous Dirichlet condition at the cavity boundary and a Neumann homogeneous condition at the outer boundary of the domain, higher integrability of the gradient of the solution is proved.

## 2. Setting of the Problem and Formulation of the Main Result

Consider a domain $\Omega \subset \mathbb{R}^d$, $d \ge 2$, with Lipschitz boundary (Lipschitz domain). Denote by $\Gamma_\varepsilon$ the hypersurface lying in $\Omega$ on the distance $\varepsilon$ from the boundary $\partial\Omega$. Here, $\varepsilon > 0$ is a small parameter. Suppose that $H_j^\varepsilon$ are balls centered on this hypersurface with radii $\alpha_j \varepsilon$, $0 < \alpha_j < \frac{1}{2}$. Denote $H_\varepsilon = \bigcup_{j \in J} H_j^\varepsilon$, $J := \{1, 2, \ldots, M_\varepsilon\}$. Here, $M_\varepsilon$ is an integer and tends to infinity as $\varepsilon \to 0$.

The domain $\Omega$ is called a Lipschitz domain, if for any point $x_0 \in \partial\Omega$ there exists an open cube $Q$ centered in $x_0$, with edges of the length $2R_0$ parallel to the coordinate axes such that $Q \cap \partial\Omega$ is a graph of the Lipschitz function $x_n = g(x_1, \ldots, x_{n-1})$ with Lipschitz constant $L$ independent of $x_0$. Here, $x = (x_1, \ldots, x_n)$ are new coordinates with origin in $x_0$.

Consider in the domain $\Omega_\varepsilon := \Omega \setminus \overline{H_\varepsilon}$, the following problem:

$$\begin{cases} \mathcal{L}u_\varepsilon = \operatorname{div} f, & \text{in } \Omega_\varepsilon, \\ u_\varepsilon = 0, & \text{on } \partial H_\varepsilon, \\ \frac{\partial u_\varepsilon}{\partial n} = 0, & \text{on } \partial\Omega, \end{cases} \tag{4}$$

where $\frac{\partial u_\varepsilon}{\partial n}$ is an outward conormal derivative of the function $u_\varepsilon$, and the components of the vector-function $f = (f_1, \ldots, f_d)$ are functions from $L_2(\Omega)$. In order to define the solution to problem (4) denoted by $W_2^1(\Omega_\varepsilon, H_\varepsilon)$, the completion of the set of infinitely smooth functions in $\overline{\Omega}$ is required, vanishing in the vicinity of $\partial H_\varepsilon$, with respect to the following norm.

$$\| u \|_{W_2^1(\Omega_\varepsilon, H_\varepsilon)} = \left( \int_{\Omega_\varepsilon} u^2 \, dx + \int_{\Omega_\varepsilon} |\nabla u|^2 \, dx \right)^{1/2}.$$

The function $u_\varepsilon \in W_2^1(\Omega_\varepsilon, H_\varepsilon)$ is called a solution to problem (4), if the following integral identity:

$$\int_{\Omega_\varepsilon} a \nabla u_\varepsilon \cdot \nabla \varphi \, dx = \int_{\Omega_\varepsilon} f \cdot \nabla \varphi \, dx, \tag{5}$$

holds for any test–function $\varphi \in W_2^1(\Omega_\varepsilon, H_\varepsilon)$ (see [12,13]).

We now study the question of higher integrability of the gradient of the solution to problem (4).

Let us describe the structure of the set $\partial H_\varepsilon$. Consider a compact set $K \subset \mathbb{R}^d$. Define the capacity $C_p(K)$ for $1 < p < d$ by the following formula.

$$C_p(K) = \inf \left\{ \int_{\mathbb{R}^d} |\nabla \varphi|^p \, dx : \ \varphi \in C_0^\infty(\mathbb{R}^d), \ \varphi \geq 1 \text{ on } K \right\}. \tag{6}$$

Let $B_r^{x_0}$ be an open ball centered in $x_0$ with radius $r$, and let $mes_{d-1}(E)$ be $(d-1)$-dimensional measure of the set $E$ (see for the definition, for instance [14]). Assume that the following is the case.

$$p = \frac{2d}{d+2}, \quad \text{as} \quad d > 2, \qquad p = \frac{3}{2}, \quad \text{as} \quad d = 2.$$

Denote by $F_s$ the boundary layer along the boundary $\partial\Omega$ with a thickness that equals to $s \geq \frac{3}{2}\varepsilon$, including all the cavities.

Suppose that for $x_0 \in F_{\frac{3}{2}\varepsilon}$ and $r \leq r_0$, either we have the following inequality:

$$C_p(\partial H_\varepsilon \cap \overline{B}_r^{x_0}) \geq c_0 r^{d-p}, \tag{7}$$

or the the following inequality:

$$mes_{d-1}(\partial H_\varepsilon \cap \overline{B}_r^{x_0}) \geq c_0 r^{d-1}, \tag{8}$$

where the positive constant $c_0$ does not depend on $x_0$ and $r$.

Note that the condition (8) is stronger but is easier to test. In addition, one can observe that under any of these conditions for any $v \in W_2^1(\Omega_\varepsilon, H_\varepsilon)$, the Friedrichs inequality of the following:

$$\int_{\Omega_\varepsilon} v^2 \, dx \leq K \int_{\Omega_\varepsilon} |\nabla v|^2 \, dx,$$

holds, which by means of the Lax–Milgram Lemma (see [15]) results in the existence of a unique solution to problem (4).

**Theorem 1.** *If $f \in L_{2+\delta_0}(\Omega)$, where $\delta_0 > 0$, then there exist positive constants $\delta(d, \delta_0) < \delta_0$ and $C$ in that the solution to problem (4) satisfies Lax-Milgra estimate:*

$$\int_{\Omega_\varepsilon} |\nabla u_\varepsilon|^{2+\delta} dx \leq C \int_{\Omega_\varepsilon} |f|^{2+\delta} \, dx, \tag{9}$$

*where $C$ depends only on $\delta_0$, $d$ and $c_0$ from (7) and (8), the constant $\lambda$ and also on $r_0 \leq R_0$ and $L$.*

### 3. Proof of the Main Result

**Proof of Theorem 1.** First of all we estimate the gradient of a solution to problem (2) in the neighbourhood of the boundary of the domain. Let us locally transform the coordinates in the vicinity of the boundary and, more precisely, in the neighbourhood of an arbitrary point $x_0 \in \partial\Omega$. By denoting the following:

$$Q_{R_0} = \{x : |x_i| < R_0, \ i = 1, \ldots, d\},$$

consider a local Cartesian coordinate system with its origin in $x_0$ and that $\partial\Omega \cap Q_{R_0}$ is given in this coordinate system by the following equation:

$$x_d = g(x'), \quad x' = (x_1, \ldots, x_{d-1}),$$

where $g$ is a Lipschitz function with the Lipschitz constant $L$. We assume that the following:

$$\Omega_{\varepsilon,R_0} = Q_{R_0} \cap \Omega_\varepsilon,$$

satisfies $x_d > g(x')$. By changing the following variables:

$$y' = x', \ y_d = x_d - g(x'),$$ (10)

we have the following:

$$P_{R_0} = \{y: \ |y_i| < R_0, \ i = 1,\ldots,d-1, \ y_d = 0\},$$

for $Q_{R_0} \cap \partial\Omega$. Denote by $\widetilde{Q}_{R_0}$ the domain $Q_{R_0}$ after the transformation of the coordinates $x \mapsto y$ (see Figure 2). $\square$

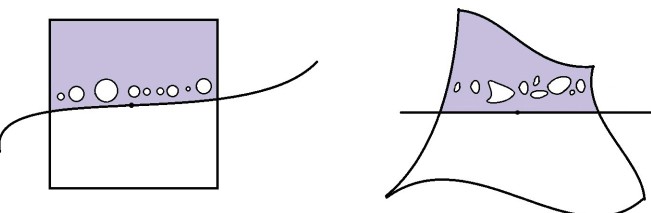

**Figure 2.** Transformation of the cube $Q_{R_0}$.

**Lemma 1.** *The domain $\widetilde{Q}_{R_0}$ contains the following cube.*

$$K_{R_0} = \{y: \ |y_i| < (1 + \sqrt{d-1}L)^{-1}R_0, \ i = 1,\ldots,d\}.$$ (11)

**Proof.** Suppose that $y \in \widetilde{Q}_{R_0}$ and $|y_i| < \vartheta R_0$ for some $\vartheta \in (0,1)$ and $i = 1,\ldots,d-1$. It is easy to observe that the following is the case:

$$y_d \in (-R_0 - g(y'), R_0 - g(y')).$$

due to the fact that function $g$ is Lipschitz and $g(0) = 0$; thus, we have the following.

$$|g(y')| \le L|y'| < \sqrt{n-1}L\vartheta R_0.$$

Consequently, the following is the case.

$$(-R_0(1 - \sqrt{n-1}L\delta), R_0(1 - \sqrt{n-1}L\delta)) \subset (-R_0 - g(y'), R_0 - g(y')).$$

Moreover, by taking the following:

$$\vartheta = \frac{1}{1 + L\sqrt{d-1}},$$

we complete the proof. $\square$

Now problem (2) in perforated semicube $K_{R_0,\text{perf}}^+ = K_{R_0} \cap \widetilde{\Omega}_\varepsilon$ has the following form.

$$\begin{cases} \widetilde{\mathcal{L}}v_\varepsilon = \operatorname{div}\widetilde{f}, & \text{in } K_{R_0,\text{perf}}^+, \\ v_\varepsilon = 0, & \text{on } \partial\widetilde{H}_\varepsilon \cap K_{R_0}, \\ \frac{\partial v_\varepsilon}{\partial \widetilde{n}} = 0, & \text{on } \partial\widetilde{\Omega} \cap K_{R_0}. \end{cases}$$ (12)

Here, $\widetilde{\Omega}$, $\widetilde{\Omega}_\varepsilon$ and $\widetilde{H}_\varepsilon$ are the images under transformation (10) of the domains $\Omega$, $\Omega_\varepsilon$ and $H_\varepsilon$, respectively, and the following:

$$\widetilde{\mathcal{L}}v := \operatorname{div}(b(y)\nabla v),$$ (13)

satisfies $b_{ij} = b_{ji}$ and

$$\mu^{-1}|\xi|^2 \leq \sum_{i,j=1}^{d} b_{ij}(y)\xi_i\xi_j \leq \mu|\xi|^2, \text{ for almost all } y \in K_{R_0}^+, \text{ and for all } \xi \in \mathbb{R}^d, \tag{14}$$

where $\mu$ depends on $\lambda$ from (1) and the constant $L$ of the function $g$. Note that the following is the case:

$$\tilde{f}(y) = (\tilde{f}_1(y), \ldots, \tilde{f}_d(y)), \text{ where } \tilde{f}_i(y) = f_i(y', y_d + g(y')), \text{ as } i = 1, \ldots, d-1,$$
$$\tilde{f}_d(y) = \sum_{i=1}^{d-1} \frac{\partial g(y')}{\partial y_i} f_i(y', y_d + g(y')) + f_d(y', y_d + g(y')), \tag{15}$$

and $\frac{\partial v_\varepsilon}{\partial \tilde{n}}$ is the respective conormal derivative.

Denote by $K_{R_0,\text{perf}}^-$ the domain $\{y : (y_1, \ldots, y_{d-1}, -y_d) \in K_{R_0,\text{perf}}^+\}$, and let $K_{R_0,\text{perf}}$ be the union $K_{R_0,\text{perf}}^- \cup K_{R_0,\text{perf}}^+$. Denote also by $\mathcal{H}_{\varepsilon,R_0}$ the cavities (pores) in $K_{R_0,\text{perf}}$ (see Figure 3).

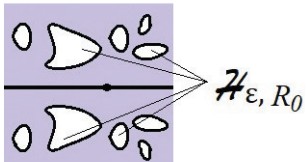

$K_{R_0,\text{perf}}$

**Figure 3.** Cube $K_{R_0,\text{perf}}$.

Let us extend the solution $v_\varepsilon$ to problem (12) by zero inside the pores and then extend it with respect to the hyperplane $\{y : y_d = 0\}$. We retain the same notation for the extended function. The extended function $v_\varepsilon$ satisfies the following problem.

$$\begin{cases} \widetilde{\mathcal{L}}_1 v_\varepsilon = \operatorname{div} h, & \text{in } K_{R_0,\text{perf}}, \\ v_\varepsilon = 0, & \text{on } \partial\mathcal{H}_{\varepsilon,R_0}. \end{cases} \tag{16}$$

Here, we have the following:

$$\widetilde{\mathcal{L}}_1 v := \operatorname{div}(c(y)\nabla v),$$

with a positive definite matrix $c = \{c_{ij}(y)\}$ satisfying $c_{jd}(y) = c_{dj}(y)$, as $j \neq d$. Moreover, $c_{jd}$ are odd extensions of the functions $b_{jd}(y)$ from (13), and $c_{ij}(y)$ are even extensions of $b_{ij}(y)$, $j \neq d$. The vector function $h = (h_1, \ldots, h_d)$ in (16) is defined by the following relations: $h_i(y)$ as $i = 1, \ldots, d-1$, are the even extension of the components $\tilde{f}_i(y)$ from (12), and $h_d(y)$ is the odd extension of $\tilde{f}_d(y)$.

Clearly the solution to problem (16) is the function $v_\varepsilon \in W_2^1(K_{R_0,\text{perf}}, \mathcal{H}_{\varepsilon,R_0})$, which satisfies the integral identity (see (5)):

$$\int_{K_{R_0,\text{perf}}} c(y)\nabla v_\varepsilon \cdot \nabla \varphi \, dy = \int_{K_{R_0,\text{perf}}} h \cdot \nabla \varphi \, dy, \tag{17}$$

for any $\varphi \in W_2^1(K_{R_0,\text{perf}}, \mathcal{H}_{\varepsilon,R_0})$. Here, $W_2^1(K_{R_0,\text{perf}}, \mathcal{H}_{\varepsilon,R_0})$ is the closure of the set of infinitely smooth functions in $K_{R_0,\text{perf}}$, vanishing in a vicinity of $\partial K_{R_0}$ and $\partial\mathcal{H}_{\varepsilon,R_0}$ by the following norm.

$$\| u \|_{W_2^1(K_{R_0,\mathrm{perf}}, \mathcal{H}_{\varepsilon,R_0})} = \left( \int\limits_{K_{R_0,\mathrm{perf}}} u^2 \, dx + \int\limits_{K_{R_0,\mathrm{perf}}} |\nabla u|^2 \, dx \right)^{1/2}.$$

We denote by $Q_R^{y_0}$ the open cube centered in $y_0$ with edges of the length $2R$ parallel to the coordinate axes. Moreover, we assume that the following is the case.

$$y_0 \in K_{\frac{R_0}{2}} \setminus \partial K_{\frac{R_0}{2}}, \text{ where } R \leq \frac{1}{2} dist(y_0, \partial K_{\frac{R_0}{2}}).$$

Denote the following:

$$\fint\limits_{Q_R^{y_0}} w \, dx = \frac{1}{|Q_R^{y_0}|} \int\limits_{Q_R^{y_0}} w \, dx,$$

where $|Q_R^{y_0}|$ is the $d$-dimensional measure of the cube $Q_R^{y_0}$.

- Consider the case $Q_{\frac{3R}{2}}^{y_0} \subset K_{R_0,\mathrm{perf}}$ and take in (17) the test-function $\varphi = (v_\varepsilon - \omega)\eta^2$, where the following is the case.

$$\omega = \fint\limits_{Q_{\frac{3R}{2}}^{y_0}} v_\varepsilon, \, dy. \tag{18}$$

Here, the cutoff function $\eta \in C_0^\infty(Q_{\frac{3R}{2}}^{y_0})$ satisfies the following.

$$0 < \eta \leq 1, \quad \eta = 1, \text{ in } Q_R^{y_0}, \quad \text{and} \quad |\nabla \eta| \leq \frac{C}{R}. \tag{19}$$

Next, the lemma is devoted to the Caccioppoli inequality.

**Lemma 2.** *For the solution $v_\varepsilon$ to problem (16), the following Caccioppoli inequality:*

$$\int\limits_{Q_R^{y_0}} |\nabla v_\varepsilon|^2 \, dy \leq C(d, \lambda, L) \left( \frac{1}{R^2} \int\limits_{Q_{\frac{3R}{2}}^{y_0}} (v_\varepsilon - \omega)^2 \, dy + \int\limits_{Q_{\frac{3R}{2}}^{y_0}} |h|^2 \, dy \right), \tag{20}$$

*holds true with $\varphi$ defined in (18).*

**Proof.** By taking $\eta$ defined in (19) and substituting the test function $\varphi = (v_\varepsilon - \omega)\eta^2$ in the integral identity (17), we have the following.

$$\int\limits_{Q_{\frac{3R}{2}}^{y_0}} c(y)|\nabla v_\varepsilon|^2 \eta^2 \, dy = -2 \int\limits_{Q_{\frac{3R}{2}}^{y_0}} c(y)\eta(v_\varepsilon - \omega)\nabla v_\varepsilon \cdot \nabla \eta \, dy + \int\limits_{Q_{\frac{3R}{2}}^{y_0}} \eta^2 h \cdot \nabla v_\varepsilon dy +$$

$$+ 2 \int\limits_{Q_{\frac{3R}{2}}^{y_0}} \eta(v_\varepsilon - \omega)h \cdot \nabla \eta \, dy. \tag{21}$$

Since $0 \leq \eta \leq 1$, then by inequality $a^2 + b^2 \geq 2ab$, we derive the following.

$$|\eta(v_\varepsilon - \varpi)\nabla v_\varepsilon \cdot \nabla \eta| \le \frac{1}{16}|\nabla v_\varepsilon|^2 \eta^2 + 4(v_\varepsilon - \varpi)^2 |\nabla \eta|^2,$$

$$|\eta^2 h \cdot \nabla v_\varepsilon| \le \frac{1}{16}|\nabla v_\varepsilon|^2 \eta^2 + 4|h|^2, \tag{22}$$

$$|\eta(v_\varepsilon - \varpi)h \cdot \nabla \eta| \le \frac{1}{16}|h|^2 + 4(v_\varepsilon - \varpi)^2 |\nabla \eta|^2.$$

Using the inequalities (21) and (22) and the ellipticity of problem (16), we obtain the following.

$$\int_{Q_{\frac{3R}{2}}^{y_0}} |\nabla v_\varepsilon|^2 \eta^2 \, dx \le C \Big( \int_{Q_{\frac{3R}{2}}^{y_0}} (v_\varepsilon - \varpi)^2 |\nabla \eta|^2 \, dx + \int_{Q_{\frac{3R}{2}}^{y_0}} |h|^2 \, dx \Big). \tag{23}$$

Finally, bearing in mind that $\eta = 1$ in $Q_R^{y_0}$ and $|\nabla \eta| \le \frac{C}{R}$, we obtain inequality (20). The lemma is proved. $\square$

Then, by using the Poincaré–Sobolev inequality:

$$\left( \fint_{Q_{\frac{3R}{2}}^{y_0}} (v_\varepsilon - \varpi)^2 \, dx \right)^{1/2} \le C(d,p)R \left( \fint_{Q_{\frac{3R}{2}}^{y_0}} |\nabla v_\varepsilon|^p \, dx \right)^{1/p},$$

with $p \ge \frac{2d}{d+2}$, we deduce from (23) the following.

$$\left( \fint_{Q_R^{y_0}} |\nabla v_\varepsilon|^2 \, dy \right)^{1/2} \le C(d,\lambda,L,p) \left( \left( \fint_{Q_{2R}^{y_0}} |\nabla v_\varepsilon|^p \, dy \right)^{1/p} + \left( \fint_{Q_{2R}^{y_0}} |h|^2 \, dy \right)^{1/2} \right). \tag{24}$$

- Consider the case $Q_{\frac{3R}{2}}^{y_0} \cap \mathcal{H}_{\varepsilon,R_0} \ne \emptyset$. Taking in (17) the test-function $\varphi = v_\varepsilon \eta^2$ with $\eta$ defined in (19), we come to (20) with $\varpi = 0$; hence, we have the following.

$$\int_{Q_R^{y_0}} |\nabla v_\varepsilon|^2 \, dy \le C(d,\lambda,L,p) \left( \frac{1}{R^2} \int_{Q_{2R}^{y_0}} v_\varepsilon^2 \, dy + \int_{Q_{2R}^{y_0}} |h|^2 \, dy \right). \tag{25}$$

Now, we estimate the first term in the right hand side of (25). If $Q_{\frac{3R}{2}}^{y_0} \cap \mathcal{H}_{\varepsilon,R_0} \ne \emptyset$, then there exists $z_0 \in Q_{\frac{3R}{2}}^{y_0} \cap \partial \mathcal{H}_{\varepsilon,R_0}$ such that $\overline{Q}_{\frac{R}{2}}^{z_0} \subset \overline{Q}_{2R}^{y_0}$. Denote by $z$ the pre-image (the inverse image) of point $z_0$ with respect to transformation (10). Note that the pre-image of the cube $\overline{Q}_{\frac{R}{2}}^{z_0}$ contains the ball $\overline{B}_{cR}^z$, with a positive constant $c$ dependent on $L$ and $d$. Due to (7), we have the following.

$$C_p(\partial \mathcal{H}_{\varepsilon,R_0} \cap \overline{B}_{cR}^z) \ge C(L,d,c_0)R^{d-p}.$$

Hence, by using the definition (6), we obtain $C_p(\partial \mathcal{H}_{\varepsilon,R_0} \cap \overline{Q}_{2R}) \ge C(L,d,c_0)R^{d-p}$. Keeping in mind the imbedding theorem (see [1] (§14.1.2)), we estimate the following.

$$\left( \fint_{Q_{2R}^{y_0}} v_\varepsilon^2 \, dy \right)^{1/2} \le C(d,p,L,c_0)R \left( \fint_{Q_{2R}^{y_0}} |\nabla v_\varepsilon|^p \, dy \right)^{1/p}. \tag{26}$$

If we use condition (8), then, bearing in mind the estimate from Proposition 4 from [1] (§13.1.1), we also obtain (26). Thus, estimate (25) results in inequality (24). Next, estimate (24) for any cubes $Q_R^{y_0}$ and the Gehring Lemma (see [16,17] and also [18] (Ch. VII)) produces the following inequality:

$$\int_{K_{\frac{R_0}{4}}} |\nabla v_\varepsilon|^{2+\delta} \, dy \le C(d, \lambda, \delta_0, c_0, L, R_0) \int_{K_{\frac{R_0}{2}}} |h|^{2+\delta} \, dy, \tag{27}$$

if $h \in L_{2+\delta_0}(K_{R_0})$, $\delta_0 > 0$, with positive constant $\delta = \delta(d, \delta_0) \le \delta_0$. Rewriting (27) and keeping in mind the properties of the extended functions, we have the following.

$$\int_{K^+_{\frac{R_0}{4}}} |\nabla v_\varepsilon|^{2+\delta} \, dy \le C(d, \lambda, \delta_0, c_0, L, R_0) \int_{K^+_{\frac{R_0}{2}}} |\widetilde{f}|^{2+\delta} \, dy. \tag{28}$$

Considering the inverse to (10) transformation, we conclude that the pre-image of $K_{\frac{R_0}{2}}$ is contained in $\Omega_{\varepsilon, R_0}$, and the pre-image of cube $K^+_{\frac{R_0}{4}}$ contains the domain $\Omega_{\varepsilon, \mu R_0}$, where $\mu = \mu(d, L) > 0$. By means of (15) and (28), we obtain the following:

$$\int_{\Omega_{\varepsilon, \mu R_0}} |\nabla u_\varepsilon|^{2+\delta} \, dx \le C(d, \lambda, \delta_0, c_0, L, R_0) \int_{\Omega_{\varepsilon, R_0}} |f|^{2+\delta} \, dx,$$

or the following.

$$\int_{\Omega_\varepsilon \cap Q^{x_0}_{\mu R_0}} |\nabla u_\varepsilon|^{2+\delta} \, dx \le C(d, \lambda, \delta_0, c_0, L, R_0) \int_{\Omega_\varepsilon \cap Q^{x_0}_{R_0}} |f|^{2+\delta} \, dx.$$

Due to the arbitrariness of point $x_0 \in \partial\Omega$ and the compactness of boundary $\partial\Omega$, one can find such finite cover of $\partial\Omega$ such that the closed set:

$$\Omega_{\varepsilon, \mu_1 R_0} = \{x \in \Omega : \ dist(x, \partial\Omega) \le \mu_1 R_0\}, \quad \mu_1 = \mu_1(d, L) > 0,$$

is contained in the union of the sets $\Omega_\varepsilon \cap Q^{x_i}_{\mu R_0}$, where $x_i \in \partial\Omega$. By summarizing the following inequalities:

$$\int_{\Omega_\varepsilon \cap Q^{x_i}_{\mu R_0}} |\nabla u_\varepsilon|^{2+\delta} \, dx \le C(d, \lambda, \delta_0, c_0, L, R_0) \int_{\Omega_\varepsilon \cap Q^{x_i}_{R_0}} |f|^{2+\delta} \, dx,$$

we derive the following.

$$\int_{F_{\mu_1 R_0}} |\nabla u_\varepsilon|^{2+\delta} \, dx \le C(d, \lambda, \delta_0, c_0, L, R_0) \int_{\Omega_\varepsilon} |f|^{2+\delta} \, dx.$$

The internal estimate of the following:

$$\int_{\Omega_\varepsilon \backslash F_{\mu_1 R_0}} |\nabla u_\varepsilon|^{2+\delta} \, dx \le C(d, \lambda, \delta_0, R_0) \int_{\Omega_\varepsilon} |f|^{2+\delta} \, dx.$$

follows from [2]. Finally, we have (9).

**Remark 1.** *Note that in the case $Q^{y_0}_{\frac{3R}{2}} \cap \mathcal{H}_{\varepsilon, R_0} \ne \varnothing$, when conditions (7) and (8) are not valid, we can modify the proof to obtain the same estimate (9). In this case, we also use Lemma 2, but instead of the Friedrichs–Sobolev inequality (26), we use the Poincaré–Sobolev inequality:*

$$\left( \fint_{Q^{y_0}_{\frac{3R}{2}}} (v_\varepsilon - \omega)^2 \, dx \right)^{1/2} \le C(d, p) R \left( \fint_{Q^{y_0}_{\frac{3R}{2}}} |\nabla v_\varepsilon|^p \, dx \right)^{1/p}, \qquad p \ge \frac{2d}{d+2},$$

*with an appropriate cutoff function.*

## 4. One Application

Let us consider the following problem:

$$\begin{cases} \mathcal{L}u_\varepsilon = \operatorname{div} f, \text{ in } \Omega_\varepsilon, \\ u_\varepsilon = 0, \text{ on } \partial H_\varepsilon, \\ \frac{\partial u_\varepsilon}{\partial n} + \varkappa u_\varepsilon = 0, \text{ on } \partial\Omega, \end{cases} \tag{29}$$

where $\varkappa$ is a constant in two-dimensional domain perforated along the boundary with the limit Robin (Fourier) problem of the following form.

$$\begin{cases} \mathcal{L}u_0 = \operatorname{div} f, \text{ in } \Omega, \\ \frac{\partial u_0}{\partial n} + \varkappa u_0 = 0, \text{ on } \partial\Omega. \end{cases} \tag{30}$$

Note that sequence $\{u_\varepsilon\}$ is uniformly bounded in the Sobolev space $W_2^1$; hence, the existence of the limit function $u_0$ is obvious. We study the rate of convergence of the solution $u_\varepsilon$ to solution $u_0$ in the Sobolev space $W_2^1$.

Assume that $(r_k, \theta)$ is the polar system of coordinates centered in $p_k^\varepsilon$ (the center of the circle $H_k^\varepsilon$). Consider the following cut-off function.

$$\psi_\varepsilon = \prod_k \psi_\varepsilon^k, \quad \psi_\varepsilon^k = \psi\left(\frac{|\ln\varepsilon|}{|\ln r_k|}\right), \quad \psi(s) = \begin{cases} 0, s \leq 1, \\ 1, s \geq 1 + \sigma. \end{cases}$$

Then, substitute the test-function $\varphi_\varepsilon = \psi_\varepsilon\varphi$, $\varphi \in W_2^1(\Omega)$ in the integral identity of problem (29).

$$\int_{\Omega_\varepsilon} a\nabla u_\varepsilon \cdot \nabla\varphi_\varepsilon \, dx + \int_{\partial\Omega} \varkappa u_\varepsilon \varphi_\varepsilon \, ds = \int_{\Omega_\varepsilon} f \cdot \nabla\varphi_\varepsilon \, dx, \tag{31}$$

In order to estimate the rate of convergence depending on $\varepsilon \to 0$, we subtract the following integral identity:

$$\int_\Omega a\nabla u_\varepsilon \cdot \nabla\varphi \, dx + \int_{\partial\Omega} \varkappa u_\varepsilon \varphi \, ds = \int_\Omega f \cdot \nabla\varphi \, dx, \qquad \varphi \in W_2^1(\Omega), \tag{32}$$

of the limit problem (30) from integral identity (31) . We obtain the following.

$$\begin{aligned} &\int_\Omega a(\psi_\varepsilon\nabla u_\varepsilon - \nabla u_0) \cdot \nabla\varphi \, dx + \int_{\partial\Omega} \varkappa(u_\varepsilon - u_0)\varphi \, ds \\ &= \int_\Omega f \cdot \nabla\varphi(\psi_\varepsilon - 1) \, dx + \int_\Omega a\nabla u_\varepsilon \cdot \nabla\psi_\varepsilon\varphi \, dx + \int_\Omega f \cdot \nabla\psi_\varepsilon\varphi \, dx. \end{aligned} \tag{33}$$

Rewriting (33) and keeping in mind the ellipticity of the operator $\mathcal{L}$ by means of the Cauchy inequality and the equivalence of the norms in the Sobolev space, we derive the following.

$$\|u_\varepsilon - u_0\|^2_{W_2^1(\Omega)} \leq C\left(\int_\Omega f \cdot \nabla\varphi(\psi_\varepsilon - 1) \, dx + \int_\Omega \nabla u_\varepsilon \cdot \nabla\psi_\varepsilon \, dx\right). \tag{34}$$

The first term in the right hand side of inequality (34) is easy to estimate (due to the Cauchy inequality) by the following.

$$K M_\varepsilon^{\frac{1}{2}} \varepsilon^{\frac{1}{1+\sigma}}.$$

Here, $M_\varepsilon$ is the number of circles, and $\varepsilon^{\frac{1}{1+\sigma}}$ is the diameter of the circle, where the integral is nontrivial, since $\psi_\varepsilon - 1 \neq 0$.

Next, we estimate the second term at the right hand side of (34) and show the difference between inequalities with and without the Meyers estimate.

1. **Without Meyers**

We have the following.

$$
\int_\Omega \nabla u_\varepsilon \cdot \nabla \psi_\varepsilon \, dx \le \left( \int_\Omega |\nabla u_\varepsilon|^2 \, dx \right)^{\frac{1}{2}} \left( \int_\Omega |\nabla \psi_\varepsilon|^2 \, dx \right)^{\frac{1}{2}}
$$

$$
\le K_1 M_\varepsilon^{\frac{1}{2}} |\ln \varepsilon| \left( \int_\varepsilon^{\varepsilon^{\frac{1}{1+\sigma}}} |\ln r|^{-4} d \ln r \right)^{\frac{1}{2}} \le K_2 M_\varepsilon^{\frac{1}{2}} |\ln \varepsilon|^{-\frac{1}{2}}.
$$

The number of circles can be the following:

$$
M_\varepsilon = |\ln \varepsilon|^{1-\chi},
$$

where constant $\chi$ satisfies $0 < \chi < 1$. In this case, we have the final estimate.

$$
\|u_\varepsilon - u_0\|_{W_2^1(\Omega)}^2 \le C |\ln \varepsilon|^{-\frac{\chi}{2}}. \tag{35}
$$

2. **With Meyers**

Suppose that the following is the case.

$$
p_1 = 2 + \delta > 2, \quad p_2 = \frac{2+\delta}{1+\delta} < 2.
$$

We obtain the following.

$$
\int_\Omega \nabla u_\varepsilon \cdot \nabla \psi_\varepsilon \, dx \le \left( \int_\Omega |\nabla u_\varepsilon|^{p_1} \, dx \right)^{\frac{1}{p_1}} \left( \int_\Omega |\nabla \psi_\varepsilon|^{p_2} \, dx \right)^{\frac{1}{p_2}}
$$

$$
\le K_1 M_\varepsilon^{\frac{1}{p_2}} \varepsilon^{\frac{2-p_2}{p_2(1+\sigma)}} |\ln \varepsilon| \left( \int_\varepsilon^{\varepsilon^{\frac{1}{1+\sigma}}} |\ln r|^{-2p_2} d \ln r \right)^{\frac{1}{p_2}} \le K_2 M_\varepsilon^{\frac{1}{p_2}} \varepsilon^{\frac{2-p_2}{p_2(1+\sigma)}} |\ln \varepsilon|^{\frac{1}{p_2}-1}.
$$

In this case, to retain the same logarithmic rate of convergence as in (35), the number of circles is as follows.

$$
M_\varepsilon = \varepsilon^{-\frac{\delta}{(1+\delta)(1+\sigma)}} |\ln \varepsilon|^{\frac{1}{1+\delta}-\chi}, \qquad 0 < \chi < \frac{1}{1+\delta},
$$

Alternatively, by keeping the logarithmic number of holes $M_\varepsilon$, we obtain the power estimate of convergence.

## 5. Discussion

Analogous results can be obtained for general perforated domains and porous media with periodic, almost periodic, nonperiodic and random structures.

## 6. Materials and Methods

In this paper, we used integral estimates of different types, Sobolev inequalities and Sobolev embedding theorems. It should be noted that the obtained inequalities (higher integrability) allowed increasing the rate of convergence and a priori estimates of solutions to homogenization problems in domains perforated along the boundary (refer to such problems with regular estimates, for example, in [10]). Similar problems with concentrated masses along the boundary can be observed in [19]. We also note recent investigations on the topic raised in paper ([20–22]).

**Funding:** This work was supported by RUSSIAN SCIENCE FOUNDATION grant number 20-11-20272.

**Institutional Review Board Statement:** Not applicable.

**Informed Consent Statement:** Not applicable.

**Data Availability Statement:** Not applicable.

**Acknowledgments:** I would very much like to thank the reviewers for their valuable and important comments, which have significantly improved the quality of the paper.

**Conflicts of Interest:** The author declares no conflict of interest.

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
