# Peer review of "The Meyers Estimates for Domains Perforated along the Boundary"

_mathematics, doi:10.3390/math9233015_

Round 1
Reviewer 1 Report
*) Some parameters have not been defined in the text. Please fill this gap.
*) Some formulas have not been numbered. Please number them all.
*) The problem studied by the author is especially interesting for the practical applications that could derive from it. For example, the study of perforated domains certainly brings to mind the problems related to the attachment of the wing to the fuselage of an aircraft. In this regard, I ask the authors to specify whether the functional space W ^ 1_2 (Ωε, Hε) is suitable for such an application by putting the following relevant work in the bibliography:
doi: 10.3390/MATH8010006
The need to include this work in the bibliography derives from the fact that during a flight, the wing-fuselage attachment steel plate of an aircraft is subjected to cyclic thermal stresses caused by the flight altitude variation that could compromise the functionality of the plate. Therefore, after a sequence of flights it is mandatory to assess the health of the plaque. In this work a new dynamic model is proposed based on the physical transmission of heat by conduction governed by a parabolic partial differential equation of the second order with suitable initial and boundary conditions to analyze and predict the thermal stresses in the plate of a P64 OSCAR B aircraft. Developing this model in the COMSOL Multi-Physics environment, a finite element technique was applied to obtain the map of the thermal stresses on the plate. The results obtained, equivalent to those obtained from a campaign of experimental infrared thermographic measurements (not yet used in the aeronautical industry), highlight the evolution of the thermal load of the wing-fuselage attachment plate, adding evidence of possible fatigue phenomena. when entering, identify in advance if the steel plate needs to be replaced.
*) Please make the captions self-explanatory.
*) Please, specify better the chain of inequalities (19).
Author Response
Dear referee,
Thank you very much for your careful reading and comments. I improved the text according to your suggestions. Added in the introductory part and in conclusions some new paragraphs. Also added the references and verified the paragraphs you pointed out.
With best regards,
Gregory A. Chechkin
Reviewer 2 Report
From my point of view, it is an interesting topic on to investigate an elliptical problem on a perforated slope along the boundary. So, assuming a homogeneous Dirichlet condition at the cavity boundary, and a Neumann homogeneous condition at the outer boundary of the domain, higher integrability of the gradient of the solution is proved. I think that the paper entitled "THE MEYERS ESTIMATES FOR DOMAINS PERFORATED ALONG THE BOUNDARY" is well-organized and the significance of the main ideas are attractive. However, it needs some revisions from the point of authors and readers to improve the quality of the paper. After these minor revisions, I suggest that this paper can be accepted to publish in "Mathematics".
My other comments are as follows:
- The English of the paper is poor and it should edited, essentially.
- The author must use “,” or “.” in the end of each formula. For instance in formula (1) div f should be div f,
- Line 1: In the paper --> In the paper,
- Line 11: were p>2 --> where p>2
- I consider that the introduction should specify the novelty of the paper compared to other papers published in this area.
- Line 13: It is better to give a reference for “was proved that the gradient of the solution is integrable in the 13 power greater than 2”.
- For non-expert readers, it is better to give a reference for line 41.
- Line 44: For non-expert readers, it is better to write the definition of mesd−1(E).
- Lines 55 and 57: were C --> where C
- Line 121: Now --> Now,
- Line 138: \xi is a constant --> where \xi is a constant
- I some formulas, such as (28), the author should use the command: \begin{align}…..\end{align}. In formula (28) an equal sign is used at the end of the first line, and immediately an equal sign is used at the beginning of the next line! A reader may wonder at that moment why these two equal signs have repeated. For the formulas after the lines 158 and 162 too (two <= were repeated).
- Also, I consider the literature is not enough and that is why, I recommend the authors to refer to other recent works indexed in Web of Science, Scopus, Emerald, Cambrige, and of course MDPI Journals. For instance:
- Anop, A.; Chepurukhina, I.; Murach, A. Elliptic Problems with Additional Unknowns in Boundary Conditions and Generalized Sobolev Spaces. Axioms 2021, 10, 292. https://doi.org/10.3390/axioms10040292.
- Motreanu, D.; Tornatore, E. Quasilinear Dirichlet Problems with Degenerated p-Laplacian and Convection Term. Mathematics 2021, 9, 139. https://doi.org/10.3390/math9020139.
- Motreanu, D.; Sciammetta, A.; Tornatore, E. A Sub-Supersolution Approach for Robin Boundary Value Problems with Full Gradient Dependence. Mathematics 2020, 8, 658. https://doi.org/10.3390/math8050658.
Briefly, I recommend publishing after doing above minor revisions.
With many thanks and best regards.
Author Response
Dear referee,
Thank you very much for your comments and suggestions. I improved a bit the English, added the text you asked, in the introductory part and in conclusions. Also added the references and verified the lines you pointed out.
With best regards,
Gregory A. Chechkin